# INSTRUCTION-FOLLOWING AGENTS WITH JOINTLY PRE-TRAINED VISION-LANGUAGE MODELS

## ABSTRACT

Humans are excellent at understanding language and vision to accomplish a wide range of tasks. In contrast, creating general instruction-following embodied agents remains a difficult challenge. Prior work that uses pure language-only models lack visual grounding, making it difficult to connect language instructions with visual observations. On the other hand, methods that use pre-trained vision-language models typically come with divided language and visual representations, requiring designing specialized network architecture to fuse them together. We propose a simple yet effective model for robots to solve instruction-following tasks in vision-based environments. Our *Instruct*RL method consists of a multimodal transformer that encodes visual observations and language instructions, and a policy transformer that predicts actions based on encoded representations. The multimodal transformer is pre-trained on millions of image-text pairs and natural language text, thereby producing generic cross-modal representations of observations and instructions. The policy transformer keeps track of the full history of observations and actions, and predicts actions autoregressively. We show that this unified transformer model outperforms all state-of-the-art pre-trained or trained-from-scratch methods in both single-task and multi-task settings. Our model also shows better model scalability and generalization ability than prior work.[1]

## 1 INTRODUCTION

Humans are able to understand language and vision to accomplish a wide range of tasks. Many tasks require language understanding and vision perception, from driving to whiteboard discussion and cooking. Humans can also generalize to new tasks by building upon knowledge acquired from previously-seen tasks. Meanwhile, creating generic instruction-following agents that can generalize to multiple tasks and environments is one of the central challenges of reinforcement learning (RL) and robotics.

Driven by significant advances in learning generic pre-trained models for language understanding (Devlin et al., 2018; Brown et al., 2020; Chowdhery et al., 2022), recent work has made great progress towards building instruction-following agents (Lynch & Sermanet, 2020; Mandlekar et al., 2021; Ahn et al., 2022; Jang et al., 2022; Guhur et al., 2022; Shridhar et al., 2022b). For example, SayCan (Ahn et al., 2022) exploits PaLM models (Chowdhery et al., 2022) to generate language descriptions of step-by-step plans from language instructions, then executes the plans by mapping the steps to predefined macro actions. HiveFormer (Guhur et al., 2022) uses a pre-trained language encoder to generalize to multiple manipulation tasks. However, a remaining challenge is that pure language-only pre-trained models are disconnected from visual representations, making it difficult to differentiate vision-related semantics such as colors. Therefore, visual semantics have to be further learned to connect language instructions and visual inputs.

Another category of methods use pre-trained vision-language models, which have shown great success in joint visual and language understanding (Radford et al., 2021). This has made tremendous progress towards creating a general RL agent (Zeng et al., 2022; Khandelwal et al., 2022; Nair et al., 2022b; Khandelwal et al., 2022; Shridhar et al., 2022a). For example, CLIPort (Shridhar et al., 2022a) uses CLIP (Radford et al., 2021) vision encoder and language encoder to solve manipulation

---

[1]The code of *Instruct*RL is available at https://sites.google.com/view/instructrl/

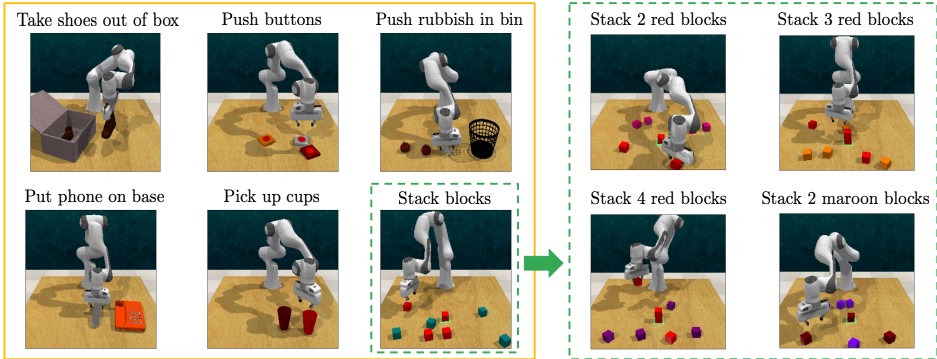

Figure 1: Examples of RLBench tasks considered in this work. **Left**: *Instruct*RL can perform multiple tasks from RLBench given language instructions, by leveraging the representations of a pre-trained vision-language transformer model, and learning a transformer policy. **Right**: Each task can be composed of multiple variations that share the same skills but differ in objects. For example, in the block stacking task, *Instruct*RL can generalize to varying colors and ordering of the blocks.

tasks. However, a drawback is that they come with limited language understanding compared to pure language-only pre-trained models like BERT (Devlin et al., 2018), lacking the ability to follow long and detailed instructions. In addition, the representations of visual input and textual input are often disjointly learned, so such methods typically require designing specialized network architectures on top of the pre-trained models to fuse them together.

To address the above challenges, we introduce *Instruct*RL, a simple yet effective method based on the multimodal transformer (Vaswani et al., 2017; Tsai et al., 2019). It first encodes fine-grained cross-modal alignment between vision and language using a pre-trained multimodal encoder (Geng et al., 2022), which is a large transformer (Vaswani et al., 2017; He et al., 2022) jointly trained on image-text (Changpinyo et al., 2021; Thomee et al., 2016) and text-only data (Devlin et al., 2018). The generic representations of each camera and instructions form a sequence, and are concatenated with the embeddings of proprioception data and actions. These tokens are fed into a multimodal policy transformer, which jointly models dependencies between the current and past observations, and cross-modal alignment between instruction and views from multiple cameras. Based on the output representations from our multimodal transformer, we predict 7-DoF actions, *i.e.*, position, rotation, and state of the gripper.

We evaluate *Instruct*RL on RLBench (James et al., 2020), measuring capabilities for single-task learning, multi-task learning, multi-variation generalization, long instructions following, and model scalability. On all 74 tasks which belong to 9 categories (see Figure 1 for example tasks), our *Instruct*RL significantly outperforms state-of-the-art models (Shridhar et al., 2022a; Guhur et al., 2022; Liu et al., 2022), demonstrating the effectiveness of joint vision-language pre-trained representations. Moreover, *Instruct*RL not only excels in following basic language instructions, but is also able to benefit from human-written long and detailed language instructions. We also demonstrate that *Instruct*RL generalizes to new instructions that represent different variations of the task that are unseen during training, and shows excellent model scalability with performance continuing to increase with larger model size.

## 2  RELATED WORK

**Language-conditioned RL with pre-trained language models.** Pre-trained language models have been shown to improve the generalization capabilities of language-conditioned agents to new instructions and to new low-level tasks (Lynch & Sermanet, 2020; Hill et al., 2020; Nair et al., 2022a; Jang et al., 2022; Ahn et al., 2022; Huang et al., 2022). Some prior work use prompt engineering with large language models (Brown et al., 2020; Chowdhery et al., 2022) to extract temporally extended plans over predefined skills (Huang et al., 2022; Ahn et al., 2022; Jiang et al., 2022), similar to work that decomposes high-level actions into sub-goals (Team et al., 2021). These work rely purely on language models to drive agents and require converting observations into language through predefined APIs. Others combine pre-trained language representations with visual inputs (Jang et al.,

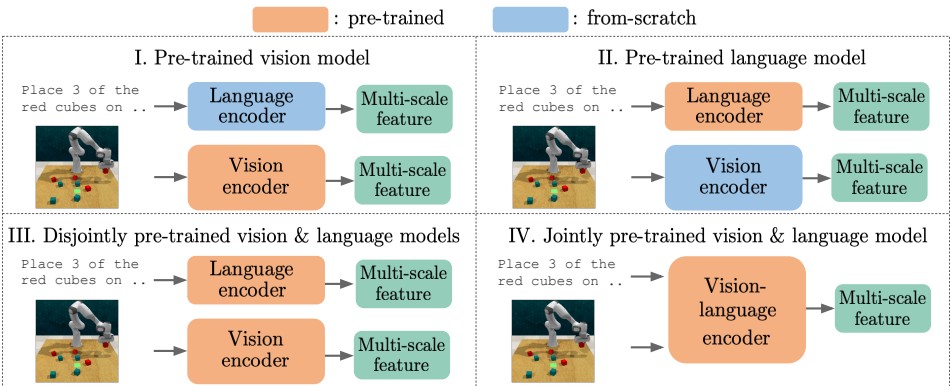

Figure 2: Different frameworks of leveraging pre-trained representations for instruction-following agents. In prior work, additional training-from-scratch is needed to combine the representations of text and image from (I) a pre-trained vision model, (II) a pre-trained language model, or (III) disjointly pre-trained language and vision models. In contrast, *Instruct*RL extracts generic representations from (IV) a jointly pre-trained vision-language model.

2022; Lynch & Sermanet, 2020; Team et al., 2021; Khandelwal et al., 2022) using specialized architectures such as UNet (Ronneberger et al., 2015) and FiLM (Perez et al., 2018). Such approaches have demonstrated success in solving challenging robotic manipulation benchmarks (Guhur et al., 2022; Shridhar et al., 2022b). In this work, we argue that using *jointly pre-trained* vision-language representations with RL can achieve superior performance in solving complex language-specified tasks, and show that our proposed approach enjoys better scalability and simpler architecture. Our work is also complementary to prompt-based methods, *e.g.*, SayCan (Ahn et al., 2022). This work focuses on improving the mapping from language instructions to robot actions, and we expect that combining our approach with prompt-based methods can achieve greater success.

**Language-conditioned RL with pre-trained vision-language models.** There have been strong interests in leveraging pre-trained vision-language models for language-conditioned RL (Shridhar et al., 2022a; Zeng et al., 2022; Khandelwal et al., 2022), motivated by the effectiveness of vision-language models such as CLIP (Radford et al., 2021). However, these methods use disentangled pipelines for visual and language input, with the language primarily being used to guide perception. Our work uses jointly pre-trained vision-language models that comes with better grounding text-to-visual content (Geng et al., 2022). The effectiveness of such jointly pre-trained models enables a simple final model, which is a jointly pre-trained vision-language transformer followed by a policy transformer. While using pretrained vision-language models has been explored in grounded navigation (Guhur et al., 2021; Hao et al., 2020; Majumdar et al., 2020; Shah et al., 2022), our work focuses on manipulation tasks which have combinatorial complexity (composition of objects/actions). Moreover, in contrast to these methods, our method *Instruct*RL is a simple architecture that scales well to large-scale tasks and can directly merge long complex language instructions with visual input.

**RL with Transformers.** Transformers (Vaswani et al., 2017) have led to significant gains in natural language processing (Devlin et al., 2018; Brown et al., 2020), computer vision (Dosovitskiy et al., 2020; He et al., 2022) and related fields (Lu et al., 2019; Radford et al., 2021; Geng et al., 2022). They have also been used in the context of supervised reinforcement learning (Chen et al., 2021a; Reed et al., 2022), vision-language navigation (Chen et al., 2021b; Shah et al., 2022), robot learning and behavior cloning from noisy demonstrations (Shafiullah et al., 2022; Cui et al., 2022), and language-conditioned RL (Guhur et al., 2022; Shridhar et al., 2022a). Inspired by their success, we leverage the transformer architecture to extract pre-trained representations from language and vision and learn a language-conditioned policy.

## 3 PROBLEM DEFINITION

We consider the problem of robotic manipulation from visual observations and natural language instructions. We assume the agent receives a natural language instruction $\mathbf{x} := \{x_1, \ldots, x_n\}$ con-

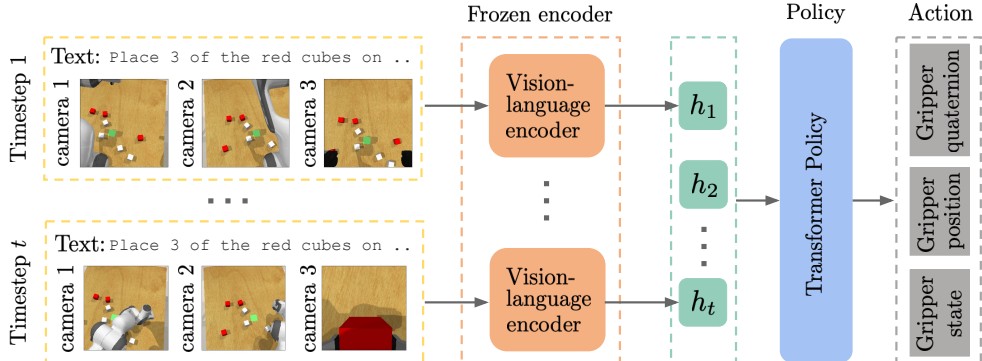

Figure 3: *Instruct*RL is composed of a vision-language transformer and a policy transformer. First, the instruction (text) and multi-view image observations are jointly encoded using the pre-trained vision-language transformer. Next, the sequence of representations and a history of actions are encoded by the policy transformer to predict the next action.

sisting of $n$ text tokens. At each timestep $t$, the agent receives a visual observation $o_t \in \mathcal{O}$ and takes an action $a_t \in \mathcal{A}$ in order to solve the task specified by the instruction.

We parameterize the policy $\pi\left(a_t \mid \mathbf{x}, \{o_i\}_{i=1}^{t}, \{a_i\}_{i=1}^{t-1}\right)$ as a transformer model, which is conditioned on the instruction $\mathbf{x}$, observations $\{o_i\}_{i=1}^{t}$, and previous actions $\{a_i\}_{i=1}^{t-1}$. For robotic control, we use macro steps (James & Davison, 2022), which are key turning points in the action trajectory where the gripper changes its state (open/close) or the joint velocities are set to near zero. Following James & Davison (2022), we employ an inverse-kinematics based controller to find a trajectory between macro-steps. In this way, the sequence length of an episode is significantly reduced from hundreds of small steps to typically less than 10 macro steps.

**Observation space**: Each observation $o_t$ consists of images $\{c_t^k\}_{k=1}^{K}$ taken from $K$ different camera viewpoints, as well as proprioception data $o_t^{\mathrm{P}} \in \mathbb{R}^4$. Each image $c_t^k$ is an RGB image of size $256 \times 256 \times 3$. We use $K = 3$ camera viewpoints located on the robot's wrist, left shoulder, and right shoulder. The proprioception data $o_t^{\mathrm{P}}$ consists of 4 scalar values: gripper open, left finger joint position, right finger joint position, and timestep of the action sequence. Note that we do not use point cloud data in order for our method to be more flexibly applied to other domains. Since RLBench consists of sparse-reward and challenging tasks, using point cloud data can benefit performance (James & Davison, 2022; Guhur et al., 2022), but we leave this as future work.

**Action space**: Following the standard setup in RLBench (James & Davison, 2022), each action $a_t := (p_t, q_t, g_t)$ consists of the desired gripper position $p_t = (x_t, y_t, z_t)$ in Cartesian coordinates and quaternion $q_t = (q_t^0, q_t^1, q_t^2, q_t^3)$ relative to the base frame, and the gripper state $g_t$ indicating whether the gripper is open or closed. An object is grasped when it is located in between the gripper's two fingers and the gripper is closing its grasp. The execution of an action is achieved by a motion planner in RLBench.

## 4 INSTRUCTRL

We propose a unified architecture for robotic tasks called *Instruct*RL, which is shown in Figure 3. It consists of two modules: a pre-trained multimodal masked autoencoder (He et al., 2022; Geng et al., 2022) to encode instructions and visual observations, and a transformer-based (Vaswani et al., 2017) policy that predicts actions. First, the feature encoding module (Sec. 4.1) generates token embeddings for language instructions $\{x_j\}_{j=1}^{n}$, observations $\{o_i\}_{i=1}^{t}$, and previous actions $\{a_i\}_{i=1}^{t-1}$. Then, given the token embeddings, the multimodal transformer policy (Sec. 4.2) learns relationships between the instruction, image observations, and the history of past observations and actions, in order to predict the next action $a_t$.

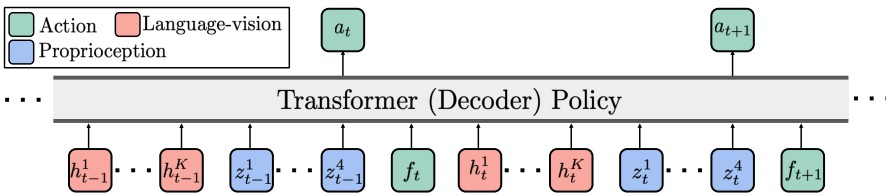

Figure 4: The architecture of the transformer policy. The model is conditioned on a history of language-vision representations and actions to predict the next action.

## 4.1 FEATURE ENCODING WITH PRE-TRAINED TRANSFORMER

We encode the instruction and visual observations using a pre-trained vision-language encoder, as shown in Figure 3. Specifically, we use a pre-trained multimodal masked autoencoder (M3AE) (Geng et al., 2022) encoder, which is a large transformer-based architecture based on ViT (Dosovitskiy et al., 2020) and BERT (Devlin et al., 2018). Specifically M3AE (Geng et al., 2022) is a transformer-based architecture that learns a unified encoder for both vision and language data via masked token prediction. It is trained on a large-scale image-text dataset(CC12M (Changpinyo et al., 2021)) and text-only corpus (Devlin et al., 2018) and is able to learn generalizable representations that transfer well to downstream tasks.

**Encoding Instructions and Observations.** Following the practice of M3AE, we first tokenize the language instructions $\{x_j\}_{j=1}^n$ into embedding vectors and then apply 1D positional encodings. We denote the language instructions as $E_x \in \mathbb{R}^{n \times d}$, where $n$ is the length of language tokens and $d$ is the embedding dimension.

We divide each image observation in $\{c_t^k\}_{k=1}^K$ into image patches, and use a linear projection to convert them to image embeddings that have the same dimension as the language embeddings. Then, we apply 2D positional encodings. Each image is represented as $E_c \in \mathbb{R}^{l_c \times d_e}$ where $l_c$ is the length of image patches tokens and $d_e$ is the embedding dimension.

The image embeddings and text embeddings are then concatenated along the sequence dimension: $E = \text{concat}(E_c, E_x) \in \mathbb{R}^{(l_c+n) \times d_e}$. The combined language and image embeddings are then processed by a series of transformer blocks to obtain the final representation $\hat{o}_t^k \in \mathbb{R}^{(l_c+n) \times d_e}$. Following the practice of VIT and M3AE, we also apply average pooling on the sequence length dimension of $\hat{o}_t^k$ to get $o_t^k \in \mathbb{R}^{d_e}$ as the final representation of the $k$-th camera image $c_t^k$ and the instruction. We use multi-scale features $h_t^k \in \mathbb{R}^d$ which are a concatenation of all intermediate layer representations, where the feature dimension $d = L * d_e$ equals the number of intermediate layers $L$ times embedding dimension $d_e$. Finally, we can get the representations over all $K$ camera viewpoints $h_t = \{h_t^1, \cdots, h_t^K\} \in \mathbb{R}^{K \times d}$ as the representation of the vision-language input.

**Encoding Proprioceptions and Actions.** The proprioception data $o_t^P \in \mathbb{R}^4$ is encoded with a linear layer to upsample the input dimension to $d$ (i.e., each scalar in $o_t^P$ is mapped to $\mathbb{R}^d$) to get a representation $z_t = \{z_t^1, \cdots, z_t^4\} \in \mathbb{R}^{4 \times d}$ all each state in $o_t^P$. Similarly, the action is projected to feature space $f_t \in \mathbb{R}^d$.

## 4.2 TRANSFORMER POLICY

We consider a context-conditional policy, which takes all encoded instructions, observations and actions as input, i.e., $\{(h_i, z_i)\}_{i=1}^t$ and $\{f_i\}_{i=1}^{t-1}$. By default, we use context length 4 throughout the paper (i.e., $4(K+5)$ embeddings are processed by the transformer policy). This enables learning relationships among views from multiple cameras, the current observations and instructions, and between the current observations and history for action prediction. The architecture of transformer policy is illustrated in Figure 4.

We pass the output embeddings of the transformer into a feature map to predict the next action $a_t = [p_t; q_t; g_t]$. We use behavioral cloning to train the models. In RLBench, we generate D, a collection of $N$ successful demonstrations for each task. Each demonstration $\delta \in$ D is composed of a sequence of (maximum) $T$ macro-steps with observations $\{o_i\}_{i=1}^T$, actions $\{a_i^*\}_{i=1}^T$ and instructions $\{x_l\}_{l=1}^n$. We minimize a loss function $\mathcal{L}$ over a batch of demonstrations B $= \{\delta_j\}_{j=1}^{|B|} \subset$ D. The

loss function is the mean-square error (MSE) on the gripper's action:

$$\mathcal{L} = \frac{1}{|B|} \sum_{\delta \in \mathsf{B}} \left[ \sum_{t \leq T} \text{MSE}\left(a_t, a_t^*\right) \right].$$ (1)

## 5 EXPERIMENTAL SETUP

To evaluate the effectiveness of our method, we run experiments on RLBench (James et al., 2020), a benchmark of robotic manipulation task (see Figure 1). We use the same setup as in Guhur et al. (2022), including the same set of 74 tasks with 100 demonstrations per task for training, and the same set of instructions for training and inference, unless stated otherwise. We group the 74 tasks into 9 categories according to their key challenges (see Appendix A.7 for each category's description and list of tasks).

For each task, there are a number of possible *variations*, such as the shapes, colors, and ordering of objects; the initial object positions; and the goal of the task. These variations are randomized at the start of each episode, during both training and evaluation. Based on the task and variation, the environment generates a natural language task instruction using a language template (see Appendix A.3).

We compare *Instruct*RL with strong baseline methods from three categories:

- *RL with pre-trained language model*: **HiveFormer** (Guhur et al., 2022) is a state-of-the-art method for instruction-following agents that uses a multimodal transformer to encode multi-camera views, point clouds, proprioception data, and language representations from a pre-trained CLIP language encoder (Radford et al., 2021). We report the published results of HiveFormer unless otherwise mentioned.

- *RL with pre-trained vision-language model*: **CLIP-RL** is inspired by CLIPort (Shridhar et al., 2022a), which demonstrates the effectiveness of CLIP for robotic manipulation. CLIP-RL uses concatenated visual- and language-representations from a pre-trained CLIP model. Similar to *Instruct*RL, CLIP-RL uses multi-scale features by concatenating intermediate layer representations, and is trained using the same hyperparameters.

- *RL trained from scratch*: **Auto-$\lambda$** (Liu et al., 2022) is a model trained from scratch that uses the UNet network (Ronneberger et al., 2015) and applies late fusion to predictions from multiple views. We report the published results of Auto-$\lambda$ unless mentioned otherwise.

*Instruct*RL uses the official pre-trained multimodal masked autoencoder (M3AE) model (Geng et al., 2022), which was jointly pre-trained on a combination of image-text datasets (Changpinyo et al., 2021; Thomee et al., 2016) and text-only corpus (Devlin et al., 2018). CLIP-RL uses the official pre-trained CLIP models. See Appendix A.2 for more details about the pre-training datasets.

All models are trained for 100K iterations. For evaluation, we measure the per-task success rate for 500 episodes per task. Further implementation and training details can be found in Appendix A.1.

## 6 EXPERIMENTAL RESULTS

We evaluate the methods on single-task performance (Figure 5), multi-task performance (Figures 6, 7), generalization to unseen instructions and variations (Figure 8a), and scalability to larger model size (Figure 8b). In all metrics, we find that *Instruct*RL outperforms state-of-the-art baselines despite being a simpler method.

**Single-task performance.** Figure 5 shows that, across all 9 categories of tasks, *Instruct*RL performs on par or better than all state-of-the-art baselines. On average, *Instruct*RL significantly outperforms prior work despite being much simpler.

**Multi-task performance.** In the multi-task setting, each model is trained on a set of tasks and then evaluated on each of the training tasks. In Figure 6, we compare multi-task performance on 10 RLBench tasks considered in HiveFormer (Guhur et al., 2022) and Auto-$\lambda$ (Liu et al., 2022), with

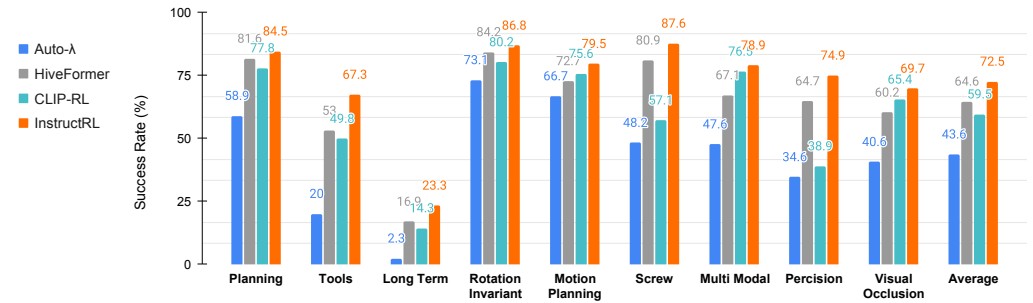

Figure 5: Single-task performance on 74 RLBench tasks from James et al. (2020); Guhur et al. (2022) grouped into 9 categories.

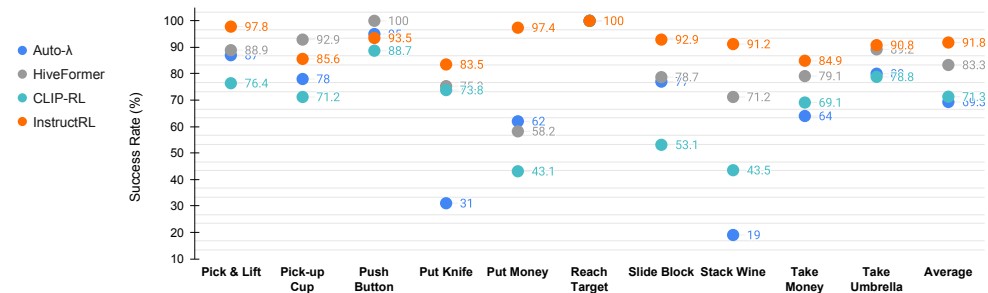

Figure 6: Multi-task performance on 10 RLBench tasks from Guhur et al. (2022); Liu et al. (2022).

100 training demonstrations per task. *Instruct*RL exhibits strong multi-task performance compared to other methods. In particular, even though both *Instruct*RL and HiveFormer use a transformer policy with language instructions, *Instruct*RL outperforms HiveFormer by a large margin. In Figure 7, we further evaluate multi-task performance on all 74 RLBench tasks, and see that *Instruct*RL outperforms CLIP-RL in most categories. These results demonstrate the transferability of jointly pre-trained vision-language models to diverse multi-task settings.

**Generalization to unseen instructions and variations.** In Figure 8a, we report the performance on the *Push Buttons* task, which requires the agent to push colored buttons in a specified sequential order given in the instruction. In this task, instructions are necessary to solve the unseen task variations correctly (*e.g.*, pushing the buttons in a blue-red-green vs. red-green-blue order cannot be inferred from only looking at the scene). We evaluate the models on instructions that are both seen and unseen during training; unseen instructions can contain new colors, or an unseen ordering of colors. We see that *Instruct*RL achieves higher performance on both seen and unseen instructions, even in the most challenging case where only 10 demonstrations are available per variation.

**Scalability to larger model size.** One of the key benefits of pre-trained models is that we can use a huge amount of pre-training data that is typically not affordable in RL and robot learning scenarios. In Figure 8b, we evaluate different model sizes on multi-task performance across 14 selected tasks (listed in Appendix A.5). For fair comparison with training-from-scratch, we fix the size of the policy transformer, and only vary the size of the transformer encoder. We compare four model sizes: B/32, B/16, L/16, and H/16, where "B" denotes ViT-base; "L" denotes ViT-large; "H" denotes ViT-huge; and "16" and "32" denote patch sizes 16 and 32, respectively.

Both CLIP-RL and *Instruct*RL improve with larger model size, but *Instruct*RL shows better scalability.[2] On the other hand, learning-from-scratch is unable to benefit from larger model capacity; in fact, a strong weight decay was needed to prevent this baseline from overfitting to the limited data.

---

[2]We only report the performances of CLIP-RL with B/32 and B/16, since the larger models (L/16 and H/16) are not provided in the open-source released models.

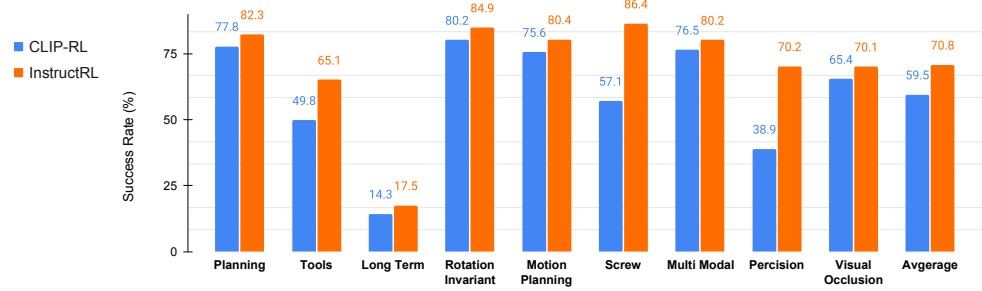

Figure 7: Multi-task performance on all 74 RLBench tasks grouped into 9 categories.

| | # Demos per Variation | Performance (Success %) | | |
|---|---|---|---|---|
| | | HiveFormer | CLIP-RL | *Instruct*RL |
| Seen Instr. | 10 | 96.8 | 95.4 | **97.8** |
| | 50 | 99.6 | 98.7 | **100.** |
| | 100 | **100.** | **100.** | **100.** |
| Unseen Instr. | 10 | 73.1 | 76.8 | **81.5** |
| | 50 | 83.3 | 84.5 | **89.4** |
| | 100 | 86.4 | 85.4 | **88.9** |

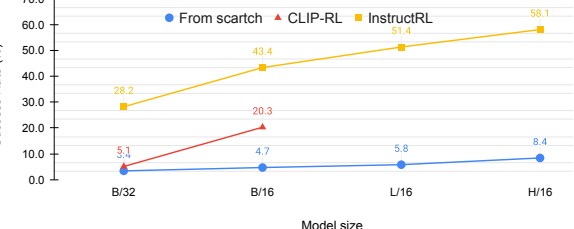

(a) Success (%) for seen and unseen instructions

(b) Multi-task performance vs. model size

Figure 8: **(a)**: Performance on the *Push Buttons* task, which has many variations on the ordering of colored buttons. *Instruct*RL achieves higher performance on both seen and unseen instructions, even with only 10 training demonstrations per variation. **(b)**: We compare four different model sizes of the transformer encoder. All methods are evaluated on multi-task performance across 14 selected tasks (listed in Appendix A.5). *Instruct*RL scales well with larger model size.

# 7 ABLATION STUDY

**Instructions are more effective with joint vision-language model.** Table 1 shows models trained with and without instructions, on 10 RLBench tasks from Guhur et al. (2022); Liu et al. (2022). While using instructions improves the performance of all methods, they are most effective with *Instruct*RL. This is not surprising, as the CLIP language encoder in HiveFormer was not pre-trained on large text corpora. One can pair CLIP with a large language model such as BERT, but doing so can often hurt performance (Guhur et al., 2022) due to the language representations not being aligned with the visual representations. On the other hand, *Instruct*RL shows the strongest performance due to using the M3AE encoder, which was jointly pre-trained on both image-text and text-only data.

**Detailed instructions require language understanding.** In Table 2, we also evaluate the methods on longer and more detailed instructions, which include step-by-step instructions specific to the task (see Table 3 for examples). The detailed instructions are automatically generated using a template that contains more details than the default instruction template from RLBench ("Default"). Each "Tuned Short Instruction" has maximum token length 77 (to be compatible with CLIP's language encoder), while each "Tuned Long Instruction" can have up to 256 tokens (to be compatible with *Instruct*RL).

We compare *Instruct*RL with CLIP-RL, which uses concatenated visual- and language-representations from CLIP, as well as a variant of CLIP-RL called BERT-RL, which uses a BERT language encoder and a trained-from-scratch vision encoder. Generally across all methods, we can see that performance increases with longer and more detailed instructions, which implies that the pre-trained language encoders can extract task-relevant information from language instructions. *Instruct*RL achieves the best performance, especially with longer instructions (*e.g.*, *Instruct*RL achieves 63.7 vs. 46.1 from BERT-RL, on the *Push Buttons* task with Tuned Long Instructions).

**Fusion strategy.** How to fuse representations from multimodal data is one of the key design choices in *Instruct*RL. In Figure 9a, we compare variants of *Instruct*RL with other fusion strategies: *Concat*

Table 1: Success rates (%) of all methods with and without instructions on 10 RLBench tasks.

| | Pick & Lift | Pick-Up Cup | Push Button | Put Knife | Put Money | Reach Target | Slide Block | Stack Wine | Take Money | Take Umbrella | Avg. |
|---|---|---|---|---|---|---|---|---|---|---|---|
| Auto-λ | 82 | 72 | 95 | 36 | 31 | 100 | 36 | 23 | 38 | 37 | 55 |
| HiveFormer | 92.2 | 77.1 | **99.6** | 69.7 | 96.2 | **100** | 95.4 | 81.9 | 82.1 | 90.1 | 88.4 |
| CLIP-RL w/o inst | 78.3 | 65.1 | 87.6 | 42 | 44.5 | 98.9 | 52.9 | 28.9 | 41.2 | 45.1 | 58.5 |
| CLIP-RL | 80.4 | 72.4 | 95.2 | 72.1 | 62.1 | **100** | 56.8 | 56.5 | 69.1 | 78.8 | 74.3 |
| InstructRL w/o inst | 75.9 | 61.2 | 85.3 | 48.9 | 48.9 | 98.7 | 54.5 | 29.5 | 43.2 | 47.8 | 59.4 |
| InstructRL | **97.8** | **84.5** | 99.5 | **84.5** | **98.7** | **100** | **97.5** | **93.2** | **89.8** | **92.94** | **93.8** |

Table 2: Comparison between using default instructions vs. longer and more detailed instructions.

| | Push Buttons | | | Light Bulb In | | |
|---|---|---|---|---|---|---|
| | Default | Tuned Short Inst | Tuned Long Inst | Default | Tuned Short Inst | Tuned Long Inst |
| BERT-RL | 41.3 | 43.1 | 46.1 | 13.6 | 12.2 | 15.9 |
| CLIP-RL | 45.4 | 51.2 | N/A | 16.4 | 20.2 | N/A |
| *Instruct*RL | **52.6** | **53.2** | **63.7** | **22.6** | **26.1** | **31.8** |

Table 3: Longer instructions for *Push Buttons* and *Stack Blocks*.

| Task | Instructions type | Example instructions |
|---|---|---|
| *Push Buttons* | Default | Push the red button, then push the green button, then push the yellow button |
| | Short Tuned Inst | Move the gripper close to red button, then push the red button, after that, move the gripper close to the green button to push the green button, finally, move the gripper close to the yellow button to push the yellow button. |
| | Long Tuned Inst | Move the white gripper closer to red button, then push red button down, after pushing red button, pull the white gripper up and move the gripper closer to green button, then push green button down, after pushing green button, pull the white gripper up and move the gripper closer to the yellow button, then push yellow button down |
| *Stack Blocks* | Default | Place 3 of the red cubes on top of each other |
| | Short Tuned Inst | Choose a red cube as the stack base, then pick another red cube and place it onto the red cube stack, repeat until the stack has 3 red cubes. |
| | Long Tuned Inst | Choose a red cube as the stack base, then pick another red cube and place it onto the red cube stack, then move the white gripper to pick another red cube and place it onto the red cube stack, repeat until the stack has 3 red cubes. |

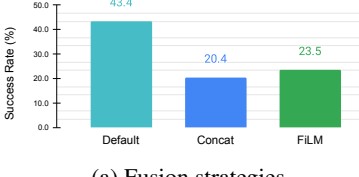

| Length | Success % |
|---|---|
| 1 | 55.1 |
| 2 | 56.2 |
| 4 | 58.1 |
| 8 | 60.3 |

| Representation selection | Success Rate |
|---|---|
| Last Layer | 47.1 |
| Second-to-Last Layer | 46.3 |
| Concat Last Eight Layers | 48.4 |
| Concat First Eight Layers | 45.1 |
| Concat All 24 Hidden Layers | **51.4** |

(a) Fusion strategies   (b) History context lengths   (c) Feature selection

Figure 9: Ablations of *Instruct*RL evaluated on the 14 selected tasks listed in Appendix A.5. We report the average success % over all tasks. **(a)**: Performance of *Instruct*RL with different strategies for fusing the language and vision features. **(b)**: Effect of history context length on the performance of *Instruct*RL. **(c)**: *Instruct*RL with features selected from different layers.

runs the encoder twice to obtain vision and language representation separately, then concatenates them together. *FiLM* (Perez et al., 2018) fuses vision and language features layer-wisely before concatenating them together. *Default* refers to *Instruct*RL where language and vision are encoded once to obtain joint representations. We see that using a joint representation of language and vision inputs is critical for performance.

**History context encoding.** The flexibility of the transformer architecture allows us to encode multiple past steps. To investigate the benefit of utilizing historical information, we ablate the history context length. Figure 9b shows that there is a steady increase in performance when we increase the context length. However, using a longer context requires more compute and memory (*e.g.*, increasing the context length from 4 to 8 adds about 20% longer training time on one task). Thus, we choose a context length of 4 for computational efficiency.

**Multi-scale features.** In Figure 9c, we compare the performance of *Instruct*RL with features selected from different layers. The results show that using a combination of intermediate representations is most effective.

## 8 CONCLUSION

In this paper, we propose *Instruct*RL, a novel transformer-based instruction following method. *Instruct*RL has a simple and scalable architecture which consists of a pre-trained multimodal transformer and a policy transformer. Experimental results show that *Instruct*RL significantly outperforms the state-of-the-art pre-trained and train-from-scratch instruction following methods. As *Instruct*RL achieves state-of-the-art results and scales well with model capacity, applying our approach to a larger scale of problems would be an interesting future work.

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

# A APPENDIX

## A.1 IMPLEMENTATION AND COMPUTE RESOURCES

We use the AdamW (Kingma & Ba, 2015; Loshchilov & Hutter, 2017) optimizer with a learning rate of $5 \times 10^{-4}$ and weight decay $5e-5$. All models are trained on TPU v3-128 using cloud TPU. Since TPU does not support headless rendering with PyRep [3] simulators that RLBench is built upon, we evaluate the models on NVIDIA Tesla V100 SXM2 GPU using headless rendering. Each training batch per device consists of 32 demonstrations sequence with length 4, in total the batch size is 512. All models are trained for 100K iterations. We apply data augmentation in training, including jitter over the RGB images $c_t^k$.

Our implementation of *Instruct*RL is built upon the official JAX implementation of multimodal masked autoencoder (M3AE) (Geng et al., 2022)[4], and we use the official pre-trained M3AE models that were pre-trained on CC12M (Changpinyo et al., 2021) and text corpus (Devlin et al., 2018).

Our implementation of CLIP-RL is built upon a JAX CLIP implementation[5], and we use the official pre-trained CLIP models[6] for the visual- and language-representations.

Both CLIP-RL and *Instruct*RL use ViT-B/16 in all experiments unless otherwise specified. As we demonstrated in experiment section, while the larger models ViT-L/16 and ViT-H/16 can further boost *Instruct*RL's results, we use ViT-B/16 to reduce computation cost and to have apple to apple comparison with baselines.

Our code is available at at `https://sites.google.com/view/instructrl/`.

## A.2 PRE-TRAINING DATASETS

### A.2.1 M3AE

M3AE is trained on a combination of image-text datasets and text-only datasets. The image-text data includes the publicly available Conceptual Caption 12M (CC12M) (Changpinyo et al., 2021), Redcaps (Desai et al., 2021), and a 15M subset of the YFCC100M (Thomee et al., 2016) selected according to (Radford et al., 2021).

The text-only data includes the publicly available Wikipedia and the Toronto BookCorpus (Zhu et al., 2015). For language data, we use the BERT tokenizer from Huggingface[7] to tokenize the text. Following Geng et al. (2022), we use 0.5 for text loss weight and 0.75 for text mask ratio. We use 0.1 for text-only loss.

### A.2.2 CLIP

CLIP is trained on the publicly available YFCC100M (Thomee et al., 2016) which consists of 100M image-text pairs.

## A.3 RLBENCH INSTRUCTIONS

We use the task instructions provided in RLBench. For each task, there are a number of possible *variations*, such as the shapes and colors of objects, the initial object positions, and the goal of the task. Based on the task and variation, the environment generates a natural language task instruction. For example, in the task *"Put groceries in cupboard"*, the generated instruction is of the form "*pick up the OBJECT and place it in the cupboard*" where OBJECT is one of 9 possible grocery items. A full list of RLBench task instructions can be found at `https://github.com/stepjam/RLBench/tree/master/rlbench/tasks`.

---

[3] `https://github.com/stepjam/PyRep`
[4] `https://github.com/young-geng/m3ae_public`
[5] `https://github.com/google-research/scenic/tree/main/scenic/projects/baselines/clip`
[6] `https://github.com/openai/CLIP`
[7] `https://huggingface.co/docs/transformers/main_classes/tokenizer`

| Task | stack blocks | | slide block | | open drawer | | push buttons | | meat off grill | | put money in safe | | stack wine | | turn tap | |
|------|------|------|------|------|------|------|------|------|------|------|------|------|------|------|------|------|
| Number of demonstrations | 100 | 10 | 100 | 10 | 100 | 10 | 100 | 10 | 100 | 10 | 100 | 10 | 100 | 10 | 100 | 10 |
| HiveFormer | 31% | 19% | 82% | 65% | 86% | 52% | 65% | 34% | 79% | 45% | 52% | 15% | 25% | 7% | 75% | 61% |
| *Instruct*RL | 41% | 32% | 89% | 78% | 94% | 81% | 89% | 61% | 92% | 65% | 75% | 42% | 37% | 12% | 83% | 68% |

Table 4: Performance (success rate %) of HiveFormer and *Instruct*RL while varying the number of demonstrations. *Instruct*RL can use fewer demonstrations to achieve high success rate, while HiveFormer requires more demonstrations.

## A.4 EVALUATION DETAILS

In multi-task setting, models are trained on multiple tasks simultaneously and subsequently evaluated on each task independently. In the generalization experiment setting, models are trained on a subset of all variations, then evaluated on unseen task variations.

## A.5 MULTI-TASK MULTI-VARIATION TASKS DETAILS

For experiments including model scaling (Figure 8b) and ablation studies (Figure 9), we selected 14 difficult tasks that have multiple variations: *Stack wine*, *Open drawer*, *Meat off grill*, *Put item in drawer*, *Turn tap*, *Put groceries in cupboard*, *Sort shape*, *Screw bulb in*, *Close jar*, *Stack blocks*, *Put item in safe*, *Insert peg*, *Stack cups*, and *Place cups*.

Each task comes with multiple variations that include objects colors, objects shapes, and the ordering of objects to be interacted with. To make a fair comparison with baselines, we use the slightly modified version of RLBench from HiveFormer (Guhur et al., 2022). The changes include adding new tasks and improving the motion planner. The full details of each task can be found in RLBench Github repo[8] and the HiveFormer Github repo[9].

## A.6 SAMPLE EFFICIENCY ABLATION

In the experiments, we used 100 demonstrations following the same setup as in prior work, and *Instruct*RL outperforms prior state-of-the-arts significantly. However, getting 100 demonstrations can be expensive and difficult in many real world tasks. We hypothesize that *Instruct*RL can achieve better sample-efficient learning thanks to the joint language-vision encoder pretrained on large-scale passive datasets.

To study this, we randomly choose a subset of 8 tasks and evaluate the performance using only 10 demonstrations. The results from Table 4 show that *Instruct*RL outperforms baselines when using 100 demonstrations or 10 demonstrations. Moreover, using only 10 demonstrations, *Instruct*RL achieves competitive or higher success rates than baselines that use 100 demonstrations. For example, on the `stack blocks` task, *Instruct*RL using 10 demos achieves 32% success rate while HiveFormer gets 31% using 100 demos. Similarly, on the `open drawer` task, *Instruct*RL using 10 demos gets 81% while HiveFormer gets 86% using 100 demos.

The results show that *Instruct*RL is more sample efficient than prior state-of-the-arts.

## A.7 TASK CATEGORIES DETAILS

To compare with baselines including Guhur et al. (2022), our experiments are conducted on the same 74 out of 106 tasks from RLBench[8]. The 74 tasks can be categorized into 9 task groups according to their key challenges as shown in Table 5.

**Task diversity.** The 74 tasks RLBench cover a wide range of challenges that are essential for robot learning, including planning over multiple sub goals (*e.g.*, picking a basket ball and then throwing the ball) and domains where there are multiple possible trajectories to solve a task due to a large affordance area of the target object (*e.g.*, the edge of a cup).

[8]https://github.com/stepjam/RLBench/tree/master/rlbench/tasks
[9]https://github.com/guhur/RLBench/tree/74427e188cf4984fe63a9c0650747a7f07434337

RLBench employs 3 keys terms: Task, Variation, and Episode. For each task there are one or more variations, and from each variation, an infinite number of episodes can be drawn. Each variation of a task comes with a list of textual descriptions that verbally summarise this variation of the task.

An example showing the distinction between task and variation is Figure 1 and Figure 4 of James et al. (2020).

**Task difficulty.** RLBench is a manipulation benchmark that is significantly more difficult than loco-motion benchmarks. Some tasks involve precise object manipulation such as 'put knife on chopping board' and 'take usb out of computer', some tasks require preceise grasping such as 'close laptop lid' and 'open and close drawer', some tasks require solving long horizon tasks that involve many composed sets of actions, for example, the 'empty dishwasher' task involves opening the washer door, sliding out the tray, grasping a plate, and then lifting the plate out of the tray. In this work we consider tasks that can be grouped into 9 categories, as shown in Table 5, to comprehensively evaluate the effectiveness of *Instruct*RL.

In addition to manipulation and perception challenges, RLBench comes with a suite of diverse task instructions that require natural language understanding. RLBench has a large and diverse vocabulary, it has over 100 content words vocabulary size (*e.g.*, table, cup, open, grasp, box, etc) with function words removed, combing with the diverse visual input and complex interactions leads to a suite of challenging instruction following manipulation tasks.

## B    INSTRUCTION TEMPLATES AND GENERATIONS

**Default instructions generation.** The default instructions templates are shown in Table 6.

**Long instructions generation.** The tuned instructions that contain step-by-step and detailed descriptions of objects are shown in Table 7.

| Group | Challenges | Tasks |
|---|---|---|
| Planning | multiple sub-goals (*e.g.*, picking a basket ball and then throwing the ball) | basketball in hoop, put rubbish in bin, meat off grill, meat on grill, change channel, tv on, tower3, push buttons, stack wine |
| Tools | a robot must grasp an object to interact with the target object | slide block to target, reach and drag, take frame off hanger, water plants, hang frame on hanger, scoop with spatula, place hanger on rack, move hanger, sweep to dustpan, take plate off colored dish rack, screw nail |
| Long term | requires more than 10 macro-steps to be completed | wipe desk, stack blocks, take shoes out of box, slide cabinet open and place cups |
| Rotation-invariant | can be solved without changes in the gripper rotation | reach target, push button, lamp on, lamp off, push buttons, pick and lift, take lid off saucepan |
| Motion planner | requires precise grasping | toilet seat down, close laptop lid, open box, open drawer, close drawer, close box, phone on base, toilet seat up, put books on bookshelf |
| Multimodal | can have multiple possible trajectories to solve a task due to a large affordance area of the target object (*e.g.*, the edge of a cup) | pick up cup, turn tap, lift numbered block, beat the buzz, stack cups |
| Precision | involves precise object manipulation | take usb out of computer, play jenga, insert onto square peg, take umbrella out of umbrella stand, insert usb in computer, straighten rope, pick and lift small, put knife on chopping board, place shape in shape sorter, take toilet roll off stand, put umbrella in umbrella stand, setup checkers |
| Screw | requires screwing an object | turn oven on, change clock, open window, open wine bottle |
| Visual-Occlusion | involves tasks with large objects and thus there are occlusions from certain views | close microwave, close fridge, close grill, open grill, unplug charger, press switch, take money out safe, open microwave, put money in safe, open door, close door, open fridge, open oven, plug charger in power supply |

Table 5: RLBench tasks used in our experiments grouped into 9 categories.

| Task | Variation Type | # Variations | Language Tempalte |
|---|---|---|---|
| basketball in hoop | NA | 1 | put the ball in the hoop |
| put rubbish in bin | NA | 1 | put rubbish in bin |
| meat off grill | Name | 2 | take the {chicken, steak} off the grill |
| meat on grill | Name | 2 | take the {chicken, steak} on the grill |
| change channel | Up / Down | 2 | turn the channel up, down |
| tv on | NA | 1 | turn on the TV |
| push buttons | Ordering & Colors | 200 | push the {color_name} button, then push the {color_name} button, then ... |
| stack wine | NA | 1 | stack wine bottle |
| slide block to target | NA | 1 | slide the block to targe |
| reach and drag | Color | 20 | use the stick to drag the cube onto the {color_name} target |
| take frame off hanger | NA | 1 | hang frame on hanger |
| water plants | NA | 1 | pick up the watering can by its handle and water the plant |
| hang frame on hanger | NA | 1 | hang frame on hanger |
| scoop with spatula | NA | 1 | take frame off hanger |
| place hanger on rack | NA | 1 | pick up the hanger and place in on the rack |
| move hanger | NA | 1 | move hanger onto the other rack |
| sweep to dustpan | NA | 1 | sweep dirt to dustpan |
| take plate off colored dish rack | Colors | 20 | put the plate between the {color_name} pillars of the dish rack |
| screw nail | NA | 1 | screw the nail in to the block |
| wipe desk | NA | 1 | wipe dirt off the desk |
| stack blocks | Number of blocks Colors. | 90 | stack blocks_to_stack {color_name} blocks |
| take shoes out of box | NA | 1 | take shoes out of box |
| slide cabinet open and place cups | Left / Right | 2 | put cup in {option} cabinet |
| reach target | Colors | 20 | reach the {color_name} target |
| push button | Colors | 18 | push the {button_color_name} |
| lamp on | NA | 1 | turn on the light |
| lamp off | NA | 1 | turn off the light |
| pick and lift | Colors | 20 | pick up the {block_color_name} block and lift it up to the target |
| take lid off saucepan | NA | 1 | take lid off the saucepan |
| toilet seat down | NA | 1 | toilet seat down |
| close laptop lid | NA | 1 | close laptop lid |
| open box | NA | 1 | open box |
| open {option} drawer | Bottom / Middle / Top | 3 | close {option} drawer |
| close {option} drawer | Bottom / Middle / Top | 3 | close option drawer |
| close box | NA | 1 | close box |
| phone on base | NA | 1 | put the phone on the base |
| toilet seat up | NA | 1 | toilet seat up |
| put books on bookshelf | Number | 3 | put {index} books on bookshelf |
| pick up cup | Colors | 20 | pick up the {target_color_name} cup |
| turn tap | Left / Right | 2 | turn {option} tap |
| lift numbered block | Number | 3 | pick up the block with the number {block_num} |
| beat the buzz | NA | 1 | beat the buzz |
| stack cups | Number of cups. Colors. | 3x30 | stack {blocks_to_stack} {color_name} cups |
| take usb out of computer | NA | 1 | take usb out of computer |
| play jenga | NA | 1 | play jenga |
| insert onto square peg | Colors | 20 | put the ring on the {color_name} spoke |
| take umbrella out of umbrella stand | NA | 1 | take umbrella out of umbrella stand |
| insert usb in computer | NA | 1 | insert usb in computer |
| straighten rope | NA | 1 | straighten rope |
| pick and lift small | Shapes ('cube', 'cylinder', 'triangular prism', 'star', 'moon') | 5 | pick up the {shape_name} and lift it up to the target |
| put knife on chopping board | NA | 1 | put the knife on the chopping board |
| place shape in shape sorter | Shapes ('cube', 'cylinder', 'triangular prism', 'star', 'moon') | 5 | put the {shape} in the shape sorter |
| take toilet roll off stand | NA | 1 | take toilet roll off stand |
| put umbrella in umbrella stand | NA | 1 | put umbrella in umbrella stand |
| setup checkers | Number | 3 | place the remaining {number} checker in its initial position on the board |
| turn oven on | NA | 1 | turn on the oven |
| change clock | NA | 1 | change the clock to show time 12.15 |
| open window | NA | 1 | open left window |
| open wine bottle | NA | 1 | open wine bottle |
| close microwave | NA | 1 | close microwave |
| close fridge | NA | 1 | close fridge |
| close grill | NA | 1 | close the grill |
| open grill | NA | 1 | open the grill |
| unplug charger | NA | 1 | unplug charger |
| press switch | NA | 1 | press switch |
| take money out safe | Number | 3 | take the money out of the {index} shelf and place it on the table |
| open microwave | NA | 1 | open microwave |
| put money in safe | Number | 3 | put the money away in the safe on the {index} shelf |
| open door | NA | 1 | open the door |
| close door | NA | 1 | close the door |
| open fridge | NA | 1 | open fridge |
| open oven | NA | 1 | open the oven |
| plug charger in power supply | NA | 1 | plug charger in power supply |

Table 6: Language instructions templates in RLBench (James et al., 2020).

| Task | Variation Type | Num of Variations | Default Language Template | Short Tuned Language Template | Long Tuned Language Template |
|---|---|---|---|---|---|
| stack blocks | Number of blocks, colors | 90 | stack {blocks_to_stack} {color_name} blocks | Move the gripper close to {1st color_name} button, then push the {2nd color_name} button, after that, move the gripper close to the {3rd color_name} button to push the {4th color_name} button, finally, ..., move the gripper close to the {n-th color_name} button to push the {n-th color_name} button. | Move the white gripper closer to {1st color_name} button, then push {2nd color_name} button down, after pushing {3rd color_name} button, pull the white gripper up and move the gripper closer to {4th color_name} button, ..., pull the white gripper up and move the gripper closer to the {nth color_name} button, then push {nth color_name} button down |
| push buttons | Ordering of buttons, colors | 200 | push {color_name} button, then push the {color_name} button, then ... | Choose a {color_name} cube as stack base, then pick another {color_name} cube and place it onto the {color_name} stack, repeat until the stack has 3 {color_name} cubes. | Choose a {color_name} cube as the stack base, then pick another {color_name} cube and place it onto the {color_name} cube stack, then move the white gripper to pick another {color_name} cube and place it onto the {color_name} cube stack, repeat until the stack has 3 {color_name} cubes. |

Table 7: Language instructions templates for tuned instructions used in Table 2 and Table 3.

