# OpenReview forum: "Instruction-Following Agents with Jointly Pre-Trained Vision-Language Models"
_ICLR.cc/2023/Conference — Submitted to ICLR 2023_

### Official Review · Reviewer_uqHe · 2022-10-22

**Confidence:** 2
**Correctness:** 3
**Technical Novelty And Significance:** 2
**Empirical Novelty And Significance:** 3
**Recommendation:** 5

**Clarity, Quality, Novelty And Reproducibility:**

**Clarity:**
Generally well-written paper, clear figures though could be improved with some legends. Some of the explanations need improvements though, see weakness section.

**Reproducibility:**
Definitely higher than many existing papers since authors provided the training code, as well as training details in the Appendix. I am missing a checkpoint for the trained model along with the code.

**Novelty:**
Not particularly novel, combines ideas from Vision-Language Transformers and Decision Transformers. It seems like the novelty is that they use an entangled vision-language representation. This is not necessarily a weakness, but I would like to see from the baselines whether the performance gains come from here or from the fact the model is pre-trained on vision and language.

**Strength And Weaknesses:**

**Strengths**

- Generally clear and well written paper. Both the contributions and the model are clearly explained, in some cases I would value a short description of the work that is cited to avoid switching to the refered paper for basic information. For instance for 4.1, it would help to explain M3AE is a transformer-based architecture via masked token prediction.

- Very simple model, reuses components trained on passive datasets which allows the model to potentially improve as these representations improve, I would assume this makes the policy train much faster than if using a transformer from scratch, but it would be great to validate that. The performance is comparable or higher than baselines in all tasks (both single and multitask settings), though as mentioned in strengths, I have some questions and suggestions regarding baselines.

- Results:
	- Nice results on unseen instructions, even though the fact that it is only in a button pussing task is slightly underwhelming. It would be amazing to see it for novel object categories as well.

	- Very good insight in Figure 4.b, in general I value the ablations performed in section 7, see weaknesses though for caveats.


**Weaknesses**
- I would include in related work methods that convert observations into language and rely purely on language models to drive agents. Note that these are different from methods such as (Huang et al. 2022), since the former group still consumes observations from the agent. Examples of those would be (Li et al. 2022, Shridhar et al. 2021)


- Clarity and Figures
	- On the flip side on clarity, I would recommend adding some more detail in the VIT policy. For instance, h_{t}^{k} are intermediate layer representations after average pooling through the sequence length dimension? Or the average pooling is applied separately on each intermediate representation? More clarity here would be appreciated. Is the linear layer for proprioception a 4x(4d) layer or 4 (1xd) layers.

	- It would help if Figure 4 had a legend for colors representing what corresponds to instr/observation, proprioception and action.

	- Figures should be more carefully made. There are overlaps between numbers in Figure 5, and Figure 6, and I see no reason why the 2 figures should have different formats. Barplots should have standard error.
	- For Fusion strategy, a figure in the appendix of the strategies would be useful, together with a figure of how is done for CLIP-RL.


- Model
	- The transformer policy is simple which is ok, but I would like a bit more motivation for it. Why not incorporating a reward at the input as done by the Decision Transformer?


- Baselines:
	- Why not including Perceiver-Actor?
	- Could authors add more detail about what is the difference between CLIP-RL and the original CLIPORT? It does seem pretty different from CLIPORT so I am not sure how it is inspired, or where is the difference with the proposed method. Is it the fact that the transformer takes separately vision and language?
	- Why do authors change the architecture in the different baselines? It is hard to know if the gains come from architectural changes or from the fact the models are trained from scratch.
	- Did authors try fine-tuning the vision-language model with the RLBench dataset/environment? Either in a MAE way to directly with the MSE objective?

- Result:
	- Long instruction settings are a great idea but they are evaluated rather underwhelmingly here. From table 2 I get that InstructRL can support more tokens than Clip-RL, but the example of instructions do not add any extra info in the task. I think the right setting or evaluation would be to include instructions that add actually new information, and see if the task is done according to those.

- It would be good to validate effect of #iter training for different baselines. I assume one advantage of the proposed method is that it learns faster.



Clarification Questions:
- It seems like the policy here is deterministic, could authros comment on what is the source of randomness when running multiple seeds? Is the environment stochastic?

- Hiveformer is not trained on large-text corpora. What is it trained on?

Minor:
- Typo: 4.2 Instruction --> instructions
- Typo Figure 5 Avgerage


Li, Shuang, et al. "Pre-trained language models for interactive decision-making." arXiv preprint arXiv:2202.01771 (2022).

Shridhar, Mohit, et al. "ALFWorld: Aligning Text and Embodied Environments for Interactive Learning." ICLR. 2021.

**Summary Of The Paper:**

This work introduces a new model for agents to perform vision and language task. The method, named InstructRL relies on a vision and language transformer (ViLT) trained on passive datasets with language and images aligned with description. This ViLT is used to convert observations and descriptions into tokens, which are consumed my a policy transformer, trained with expert demonstrations to generate gripper actions. The method is evaluated in RLBench, showing SOTA performance amongst selected baselines, generalizing to longer instructions and zero-shot task specifications.

**Summary Of The Review:**

I really value the use of pre-trained models and simple architectures for visual-language embodied tasks. The fact that the performance scales well with larger model size is promising, and it would also be interesting to see if it scales with pre-training size. The proposed model also obtains better performance than the proposed baselines. Despite the above strengths, I think the paper needs to clarify a few-points, particularly the omission of certain baselines, the effect of model architecture vs pre-training data, the choice of the policy network, and effect of fine-tuning on the RLBench dataset. If authors can clarify why the baselines I mention in my weaknesses are not mentioned, as well as my questions regarding some of the differences in the baselines selected, and correct some of the mistakes I pointed I would be happy to raise my score.

---

> ### Author Response · Authors · 2022-11-17
> **Response to reviewer uqHe (1/2)**
>
> We are happy that the reviewer finds our ideas simple and effective. We also thank the reviewer for the detailed and valuable feedback, and address the reviewer's questions below.
>
>
> > Difference between CLIP-RL and CLIPort.
>
>
> The main differences are the following:
> (a) CLIPort utilizes point cloud observations. For fair comparison, neither CLIP-RL nor our method uses point cloud data as input.
> (b) CLIP-RL uses pre-trained CLIP VIT encoders, while CLIPort uses pre-trained CLIP ResNet.
>
>
>
> > Why no comparison with Perceiver-actor
>
>
> Perceiver-Actor uses extra privileged information such as voxel, while InstructRL uses only images, making fair comparison difficult.  In addition, the source code of Perceiver-Actor is not publicly available, so we were not able to evaluate Perceiver-Actor without the extra privileged information.
>
> As an alternative, we considered HiverFormer as our baseline for the following reasons:
> (a) HiveFormer is evaluated on a more diverse and challenging 74 tasks while Perceiver-Actor is only evaluated on 18 tasks.
> (b) HiveFormer and Perceiver-Actor have a similar architecture using transformer-based encoder and policy.
>
> As InstructRL significantly outperforms HiveFormer on a diverse wide range of 74 tasks, we believe that the evaluation is comprehensive and leave comparing with Perceiver-Actor as future work.
>
>
>
>
> > Hiveformer is not trained on large-text corpora. What is it trained on?
>
>
> In HiveFormer, the instructions are encoded using a pre-trained CLIP language encoder, similar to CLIP-RL and CLIPort. The CLIP model is trained on a large dataset of image-text pairs. Unlike M3AE which our InstructRL is based on, CLIP cannot be trained on large-text corpora because it requires paired image-text data. We believe this difference helps our method to generalize well to complex, longer instructions.
>
>
>
>
> > Fine-tuning pre-trained vision-language models on RLBench
>
>
> We thank the reviewer for this suggestion. In our initial experiments, we found that fine-tuning our pre-trained vision-language model on RLBench could further improve the results. However, for fair comparison with prior methods that utilized frozen pre-trained encoders, we opted to not use fine-tuning in our experiments.
>
>
>
> > How intermediate layer representations of VIT encoder are combined before feeding into the policy network.
>
>
> The intermediate layer representations are concatenated along the feature dimension, therefore the linear layer for predicting proprioception is 4x(4d) size.
>
>
>
> > It is hard to know if the gains come from architectural changes or from pretraining.
>
>
> We think the gains are mainly from pre-training since similar model architecture was used for both baselines (HiveFormer and CLIP-RL) and our InstructRL. It is a bit hard to exactly match the model architecture due to differences in the methods but we tried to use similar model architecture (and parameters) as much as possible. However, there are big differences in pre-training:
>
> - HiveFormer: CLIP text encoder + trained-from-scratch image and point cloud encoder
> - CLIP-RL: CLIP text encoder + CLIP image encoder
> - BERT-RL: BERT text encoder + trained-from-scratch image encoder
> - Instruct-RL: multimodal MAE encoder for both image and text
>
> The CLIP model is pretrained on 100 million image-text pairs; the BERT text encoder is trained on a large text-only corpus; and multimodal MAE (ours) is pretrained on fewer 10.5 million image-text pairs and BERT’s text-only corpus. We believe our empirical comparison against HiveFormer, BERT-RL and CLIP-RL is comprehensive, covering architecture difference, pretraining effect, and data difference. Since Instruct-RL significantly outperforms all baselines on multi-task, single-task, and generalization to long instructions, the results confirm that the gains come from the jointly pretrained image-language architecture.
>
>
> > Long instruction settings are a great idea but they are evaluated rather underwhelmingly here.
>
>
> In this work, we focus on providing more detailed descriptions of objects, goals, and step-by-step instructions. We thank the reviewer for suggesting providing completely new information in the long instructions. We believe that adding such information would help learning and it will be an interesting future work. We hope our results on long instructions could motivate further research on more advanced instruction-following robotics tasks.

---

> > ### Author Response · Authors · 2022-11-17
> > **Response to reviewer uqHe (2/2)**
> >
> > > Does the environment have stochasticity?
> >
> >
> > Yes, at the beginning of each episode, the initial positions of robot arms and the locations, colors, and orientations of objects are randomly initialized. Therefore we follow prior work in averaging the results from three random seeds.
> > Following the standard setup in RLBench, our policy is deterministic. The predicted action consists of the desired gripper position in Cartesian coordinates and quaternion relative to the base frame, and the gripper state indicating whether the gripper is open or closed. The execution of an action is achieved by a motion planner. This is specified in Section 3 "action space" paragraph.
> >
> >
> >
> > > Incorporating reward in policy network
> >
> >
> > Incorporating reward in the policy network is interesting and we believe that InstructRL can perform better with reward signals as input. However, in this work, we focus on imitation learning where rewards are not available since we consider complex manipulation tasks from RLBench where well-shaped reward functions are not available.
> >
> >
> >
> > > Add a short description of the work that is cited to avoid switching to the refereed paper for basic information. For instance for 4.1, it would help to explain M3AE is a transformer-based architecture via masked token prediction.
> >
> >
> > Following the reviewer's suggestion, we have added descriptions of refereed papers, in the revision.
> >
> >
> >
> > > Include in related work methods that convert observations into language and rely purely on language models to drive agents
> >
> >
> > We have included discussion on works that use purely language models in Related Work (Section 2)in the revision.
> >
> >
> >
> > > Very simple model, reuses components trained on passive datasets which allows the model to potentially improve as these representations improve, I would assume this makes the policy train much faster than if using a transformer from scratch, but it would be great to validate that.
> >
> >
> > Yes indeed, we found that our pretrained model takes 3 times fewer iterations to reach the performance of training from scratch (30k vs 100k, averaged over 74 tasks). This is similar to how pretrained BERT or MAE models outperforms training from scratch models in NLP and computer vision.
> >
> > We have also added an additional ablation study in Appendix A.6, where we vary the number of demonstrations. The results show that our method can learn with much fewer demonstrations. In fact, using only 10 demos, our method achieves comparable or higher performance than baselines that use 100 demos (see Table 4 in Appendix).
> >
> >
> >
> > > Nice results on unseen instructions, even though the fact that it is only in a button pushing task is slightly underwhelming. It would be amazing to see it for novel object categories as well.
> >
> >
> > The PushButtons task has high combinatorial complexity, containing 200 different variations of colors, shapes, and ordering of the objects (see Table 4 and 5 in the Appendix). Higher success rates on unseen instructions shows generalization of visual perception, language understanding, and robotic control (to unseen colors, shapes, and orderings of objects). Moreover, we agree that extending the experiments to zero-shot generalization to novel object categories would be very interesting, but this requires an environment with a much more diverse set of objects, which RLBench does not support. We leave this investigation as an exciting future work.
> >
> >
> >
> >
> > > Minor edits
> >
> >
> > Thank you for the detailed feedback! We’ve updated Figure 4 by adding a legend for color.
> >
> >
> > We hope that our response addresses all of your questions, but if not, please let us know.

---

### Official Review · Reviewer_boPj · 2022-10-24

**Confidence:** 4
**Correctness:** 3
**Technical Novelty And Significance:** 3
**Empirical Novelty And Significance:** 3
**Recommendation:** 6

**Clarity, Quality, Novelty And Reproducibility:**

it is a decent (or borderline) work to show the pretrained vision-and-language transformer as encoder for instruction-following agents in the robotics tasks/environments; However, it looks the proposed model may not novel in the vision-and-language navigation tasks [1, 2, 3, 4], it is a good practice to apply the similar idea in the robotics tasks/settings.

[1]. Guhur et al., Airbert: In-domain pretraining for vision-and-language navigation, CVPR 2021

[2]. Shah et al., Lm-nav: Robotic navigation with large pre-trained models of language, vision, and action, 2022

[3]. Majumdar et al., Improving vision-and-language navigation with image-text pairs from the web, 2022

[4]. Towards learning a generic agent for vision-and-language navigation via pre-training, CVPR 2020

**Strength And Weaknesses:**

Strengths:
1. This work leverages the pretrained vision-and-language transformer as encoder for instruction-following agents;
2. This work shows promising (SoTA) results on a set of robotics benchmark.

Weakness:
1. Somehow the proposed model looks not novel in the vision-and-language navigation tasks, but may be a good attempt in the robotics tasks.

**Summary Of The Paper:**

This paper tackles the instruction-following problem, prior work uses language-only models,which lacks the grounding between language and observations, this paper proposes a method (InstructRL), which contains a vision-and-language encoder, and a policy transformer for action prediction, the vision-and-language transformer encoder is pretrained on millions of image-text pairs and natural language text; empirical experiments on a robotics benchmark show the SoTA results.

**Summary Of The Review:**

It is a borderline, empirical work to show the pretrained vision-and-language transformer as encoder for instruction-following agents in the robotics tasks/environments, and achieves the SoTA on a set of robotics benchmark; the main concern is the novelty of the proposed model/method, which was explored a lot in the vision-and-language navigation tasks.

---

> ### Author Response · Authors · 2022-11-17
> **Response to reviewer boPj**
>
> We thank the reviewer for finding our ideas to be promising and effective. At the same time, the reviewer had concerns about novelty regarding prior approaches in navigation tasks, which we address below.
>
>
> > Novelty regarding prior approaches in navigation tasks
>
>
> In this work we focus on a highly diverse suite of 74  instruction-following visual manipulation tasks. Our evaluation setup has comparable visual and language complexity to that of grounded navigation tasks.  In particular, we highlight the difficulty of RLBench tasks:
> - RLBench tasks have combinatorial complexity (from the composition of objects/actions). The set of instructions contain over 100 vocabulary words, and covers a wide range of everyday objects. During evaluation, we randomly initialize the object colors, shapes, positions, and other factors of variation at the start of each episode. Each task can contain up to 200 different object types, colors, and textures (details are included in Appendix B).
> - Manipulation tasks have a high-dimensional continuous control action space that includes euclidean position, quaternion rotation, gripper control, and precise manipulation. In comparison, grounded navigation tasks such as VirtualHome[5, 6] have predefined discrete high-level actions [1, 2, 3, 4] that only require turning direction.
>
> Thus, our evaluation setup is very challenging, and requires the model to learn both visual and language understanding as well as continuous control. For example, the "stack blocks" task requires the robot to understand the color, number, and ordering of blocks in the instruction, and then learn a continuous-control policy to pick up correct blocks and stack them in the correct order.
>
> InstructRL is simple yet significantly outperforms more complex prior state-of-the-art methods (e.g. HiveFormer and CLIP-RL) on these challenging tasks.
>
> Compared to the prior work on grounded navigation [1, 2, 3, 4], InstructRL has the following novel properties:
> Simplicity: No need for specialized architecture or downstream finetuning to merge image features and language features.
> Can directly merge long complex language instructions with visual input.
>
> Given the strong empirical results, we agree with the reviewer that it is a promising future work to apply InstructRL to navigation tasks.
>
>
> [1]. Guhur, Pierre-Louis, et al. "Airbert: In-domain pretraining for vision-and-language navigation." Proceedings of the IEEE/CVF International Conference on Computer Vision. 2021.
>
> [2]. Shah, Dhruv, et al. "Lm-nav: Robotic navigation with large pre-trained models of language, vision, and action." arXiv preprint arXiv:2207.04429 (2022).
>
> [3]. Majumdar, Arjun, et al. "Improving vision-and-language navigation with image-text pairs from the web." European Conference on Computer Vision. Springer, Cham, 2020.
>
> [4]. Hao, Weituo, et al. "Towards learning a generic agent for vision-and-language navigation via pre-training." Proceedings of the IEEE/CVF Conference on Computer Vision and Pattern Recognition. 2020.
>
> [5]. Huang, Wenlong, et al. "Language models as zero-shot planners: Extracting actionable knowledge for embodied agents." arXiv preprint arXiv:2201.07207 (2022).
>
> [6]. Li, Shuang, et al. "Pre-trained language models for interactive decision-making." arXiv preprint arXiv:2202.01771 (2022).

---

### Official Review · Reviewer_uxTU · 2022-10-25

**Confidence:** 4
**Correctness:** 4
**Technical Novelty And Significance:** 2
**Empirical Novelty And Significance:** 2
**Recommendation:** 5

**Clarity, Quality, Novelty And Reproducibility:**

The paper was very well written and was very easy to follow. The authors mentioned that the code will be released. The paper has a novelty issue, as the architecture itself seems derivative, given the HiveFormer framework.

**Strength And Weaknesses:**

Pros:
Proposed method is simple and seems easy to implement with the provided details
for the task.
Strong performance improvements across all categories on the RLBench
benchmark. Additional analyses also reveal the importance of multiple design
choices that have been used by the authors in InstructRL.
Cons:
While the proposed system is simple, there is a slight novelty issue. Stronger pretraining models are expected to provide better representations and hence improve
model performance. The architecture itself seems derivative, given the HiveFormer
framework.
In the ablation studies, InstructRL performs with longer instructions than CLIP- or
BERT-based models. However, there is no mention of what these instructions look
like and what aspect of the original instructions have been expanded to generate
InstructRL 2
the longer instructions. Without such details, it is hard to judge the pure numbers
provided for this ablation.
In the ablation wrt context length, the author(s) mention that “improvement saturates
around 4”. From the numbers in Table 9b, it is hard to arrive at this conclusion as
there is a steady increase in performance from 1 → 2 → 4 → 8.
The writing of the paper is also highly reliant on readers having prior knowledge
about the RLBench benchmark, and does not provide adequate details about the
benchmark for new readers to understand and evaluate the findings. Without
appropriate discussion, it is hard to how hard the tasks are.

Minor Edits:
Figure 5b has some spelling mistakes in the x-axis.
In Section 4.1, under ‘Encoding instructions and observations’, the final representation
of the transformer blocks is mentioned to have dimensionality ‘d’. If that’s the case,
shouldn’t the dimensionality of be where L is the number of concatenated
layer representations?

**Summary Of The Paper:**

This paper proposes a simple method to solve the task of instruction-following in multimodal environments. While recent work makes use of pre-trained transformers, their
performance is limited by (i) lack of grounding (in the case where separate vision and
language models are used) and (ii) lack of ability to follow long instructions (in the case
of CLIP). To overcome these limitations, this paper proposes using M3AE, a recently
proposed multi-modal transformer, as the backbone to provide vision-language
representations. In experiments on RLBench, InstructRL, the proposed method, is able
to outperform prior work such as HiveFormer (which makes use of separate vision and
language models) and a CLIPort-inspired CLIP-RL method across all categories of
tasks. Further analyses reveal that the model is also capable of handling longer
instructions, novel instruction combinations, and scales well with model size.


**Summary Of The Review:**

Current decision: Borderline Reject
The novelty and the lack of self-contained information in this paper imply that the paper
currently stands as a reject. However, the system proposed is very simple and hence
the decision is at borderline reject. If the author(s) are able to address the issues wrt
ablations and writing, the score can be reconsidered.

---

> ### Author Response · Authors · 2022-11-17
> **Response to reviewer uxTU**
>
> We thank the reviewer for the helpful and constructive feedback, which we address below and in our revision. We are also happy that reviewer uxTU finds our ideas to be well-motivated, simple, and effective.
>
> >  Comparison with HiveFormer, and questions about novelty
>
> As you pointed out, our method achieves considerable performance improvement over HiveFormer by 2.6% to 14% higher success rate on single-task performance over 74 tasks (Figure 5); 4.2% to 31.3% higher success rate on multi-task performance over 9 categories (Figure 6); and 3.2% to 5.2% higher success rate on generalization to unseen instructions (Figure 8). We believe these state-of-the-art results provide a notable empirical contribution to the community, showcasing the importance of a jointly pre-trained language-vision encoder for instruction-following visual robotic manipulation.
>
> Despite the performance gain, our method is actually even simpler than HiveFormer. HiveFormer consists of a pretrained language model and a separate vision model whose features have to be merged through downstream finetuning. This requires a specialized architecture design, and also requires more data from downstream robotics tasks, which may not be effective since data is highly limited in robotics. In contrast, InstructRL enjoys a simpler and more data-efficient architecture owing to the multimodal representation of the unified encoder for both language and vision. Despite its simplicity, InstructRL unlocks a higher level of understanding longer and detailed instructions, and  achieves new state-of-the-art on a wide range of tasks.
>
> > Clarification about longer instructions used in the experiments
>
> In Table 3, we list examples of original and longer instructions used in our experiments. Additionally, per the reviewer's feedback, we have added more details in Appendix B about the generation process of the longer instructions. Compared with the default instructions provided by RLBench, the longer instructions contain more descriptions about the task, objects and step-by-step instructions.
>
> > Provide adequate details about the RLBench benchmark
>
> We list the tasks according to their required skills (e.g. planning, precision, etc) in Table 4 and list the exact characterization of all 74 tasks used in this paper in Table 5.  The language instructions are reported in Table 5 and Table 6 in the Appendix.
>
> We also highlight the difficulty of RLBench tasks:
> - RLBench tasks have combinatorial complexity (from the composition of objects/actions). The set of instructions contain over 100 vocabulary words, and covers a wide range of everyday objects. During evaluation, we randomly initialize the object colors, shapes, positions, and other factors of variation at the start of each episode. Each task can contain up to 200 different object types, colors, and textures (details are included in Appendix B).
> - Manipulation tasks have a high-dimensional continuous control action space that includes euclidean position, quaternion rotation, gripper control, and precise manipulation. In comparison, grounded navigation tasks such as VirtualHome[1, 2] have predefined discrete high-level actions [3, 4] that only require turning direction.
>
> Thus, our evaluation setup is very challenging, and requires the model to learn both visual and language understanding as well as continuous control. For example, the "stack blocks" task requires the robot to understand the color, number, and ordering of blocks in the instruction, and then learn a continuous-control policy to pick up correct blocks and stack them in the correct order. Our method outperforms prior state-of-the-art significantly, even on these difficult task categories, as shown in Figure 5 and Figure 7.
>
>
> [1] Huang, Wenlong, et al. "Language models as zero-shot planners: Extracting actionable knowledge for embodied agents." arXiv preprint arXiv:2201.07207 (2022).
>
> [2] Li, Shuang, et al. "Pre-trained language models for interactive decision-making." arXiv preprint arXiv:2202.01771 (2022).
>
> [3] Guhur, Pierre-Louis, et al. "Airbert: In-domain pretraining for vision-and-language navigation." Proceedings of the IEEE/CVF International Conference on Computer Vision. 2021.
>
> [4]. Shah et al., Lm-nav: Robotic navigation with large pre-trained models of language, vision, and action, 2022
>
>
>
> > Justification on the context length
>
>
> Thank you for pointing out this detail. Indeed, there is a steady increase in performance when we increase the context length. We choose the context length of 4 due to computational efficiency since longer context is slow to compute and memory intensive. We found that increasing the context length from 4 to 8 adds about 20% longer training time on one task. We have revised the draft to clarify this (Section 7).
>
>
>
> > Minor Edits
>
> We have addressed the typos and spelling mistakes in the revision. Thanks for your suggestions.
>
>
> We hope that our response addresses all of your questions, but if not, please let us know.

---

### Author Response · Authors · 2022-11-17
**Response to all reviewers**

We thank all reviewers for their constructive comments and suggestions. We are happy to see that all reviewers generally appreciate the simplicity and effectiveness of our proposed method:
- “simple and easy to implement” (reviewers uxTU, uqHe)
- “strong performance improvements across all categories on the RLBench benchmark” (reviewer uxTU)
- "shows promising (SoTA) results on a set of robotics benchmark” (reviewer boPj)
- “The fact that the performance scales well with larger model size is promising”, “generalizing to longer instructions and zero-shot task specifications'' (reviewer uqHe).


We believe our work provides a notable contribution to the community, showcasing the importance of a jointly pre-trained language-vision encoder for instruction-following visual robotic manipulation. Our simple method achieves state-of-the-art results on a wide range of challenging RLBench tasks, which require both visual and language understanding as well as precise robotic control. In contrast to the previous state-of-the-art HiveFormer, which requires downstream finetuning to merge its separate language and vision embeddings, our method enjoys a simpler, data-efficient architecture owing to the multimodal representation of the unified encoder. Data efficiency is especially desirable since data is often highly limited in robotics. Despite its simplicity, InstructRL unlocks a higher level of understanding longer and detailed instructions, and  achieves new state-of-the-art on a wide range of tasks.

---

### Decision · Program_Chairs · 2023-01-20

**Decision:**

Reject

**Justification For Why Not Higher Score:**

There is clear consensus that the novelty is limited

**Justification For Why Not Lower Score:**

N/A

**Metareview: Summary, Strengths And Weaknesses:**

The paper considers the problem of instruction-following for tasks that require reasoning over visual inputs in addition to text (i.e., grounding). The paper proposes InstructRL, an architecture that uses a pre-trained vision-language transformer as a joint encoder, the output of which is provided to a policy transformer that predicts the agent's actions. Experiments on the RLBench benchmark demonstrate that InstructRL outperforms contemporary baselines, including HiveFormer (which employs separate vision and language encoders) and a CLIPort-inspired method. The paper additionally analyzes the method's ability to handle longer instructions as well as novel instructions in a zero-shot manner.

The use of transformer architectures for instruction-following has attracted significant interest in the machine learning community of-late. As the paper points out, instruction-following for embodied agents requires relating (grounding) textual instructions to the agent's environment. All three reviewers agree that encoding vision and language input via a transformer that is pre-trained on image-text data is sensible, and shows strong performance compared to contemporary methods in spite of the relatively simple nature of the InstructRL architecture. Because of this simplicity and the promise to release source code, the results can be readily reproduced. The primary concern shared by each reviewer is in regards to the significance of the contribution, which is primarily an empirical demonstration of the benefits of using a pre-trained vision-language encoder. The reviewers emphasize that the novelty of the architecture itself is rather limited considering the large body of work in this area. This is not to suggest that the simplicity of the InstructRL is not an advantage, but rather that the significance of this contribution needs to be made clearer.

The AC notes that none of the three reviewers engaged in a discussion with the authors or acknowledged that they had read the authors' response, despite the efforts of the AC and the authors. However, given the consensus among the reviewers together with the fact that there are not any questions about the factual accuracy of the concerns that they raise, the AC feels comfortable with the decision.

**Summary Of Ac-Reviewer Meeting:**

N/A